# Large-scale remodeling of a repressed exon ribonucleoprotein to an exon definition complex active for splicing

Somsakul Pop Wongpalee[1], Ajay Vashisht[2], Shalini Sharma[3], Darryl Chui[1], James A Wohlschlegel[2], Douglas L Black[1]*

[1]Department of Microbiology, Immunology and Molecular Genetics, University of California, Los Angeles, Los Angeles, United States; [2]Department of Biological Chemistry, University of California, Los Angeles, Los Angeles, United States; [3]Department of Basic Medical Sciences, University of Arizona, Phoenix, United States

**Abstract** Polypyrimidine-tract binding protein PTBP1 can repress splicing during the exon definition phase of spliceosome assembly, but the assembly steps leading to an exon definition complex (EDC) and how PTBP1 might modulate them are not clear. We found that PTBP1 binding in the flanking introns allowed normal U2AF and U1 snRNP binding to the target exon splice sites but blocked U2 snRNP assembly in HeLa nuclear extract. Characterizing a purified PTBP1-repressed complex, as well as an active early complex and the final EDC by SILAC-MS, we identified extensive PTBP1-modulated changes in exon RNP composition. The active early complex formed in the absence of PTBP1 proceeded to assemble an EDC with the eviction of hnRNP proteins, the late recruitment of SR proteins, and binding of the U2 snRNP. These results demonstrate that during early stages of splicing, exon RNP complexes are highly dynamic with many proteins failing to bind during PTBP1 arrest.

*For correspondence: dougb@ microbio.ucla.edu

## Introduction

Alternative pre-mRNA splicing is a key regulatory process that determines the output of most genes of higher eukaryotes (*Pan et al., 2008*; *Lee and Rio, 2015*; *Merkin et al., 2012*; *Barbosa-Morais et al., 2012*; *Fu and Ares, 2014*). Splicing choices are directed by proteins of multiple classes and structural families that bind to the nascent pre-mRNA to alter spliceosome assembly (*Lee and Rio, 2015*; *Fu and Ares, 2014*). One family of related regulators is the serine-arginine-rich proteins (SR proteins), which usually bind within an exon to stimulate its splicing (*Fu and Ares, 2014*; *Long and Caceres, 2009*; *Busch and Hertel, 2012*). Other proteins are more generally classed as the heterogeneous nuclear ribonucleoproteins (hnRNP proteins) that belong to multiple structural families and bind to specific sites throughout the nascent pre-mRNA (*Busch and Hertel, 2012*; *Dreyfuss et al., 1993*). Depending on the family and binding site location, hnRNP proteins can act to either repress or stimulate splicing of particular exons or splice sites (*Fu and Ares, 2014*). The interactions of these pre-mRNA-bound regulatory proteins with the general splicing apparatus or spliceosome are largely unknown.

The pathway of spliceosome assembly has been determined for introns of yeast and for several unusually short mammalian introns (*Will and Lührmann, 2011*; *Wahl et al., 2009*). An early step in spliceosome assembly across a short intron is the formation of the prespliceosome or A complex that contains the U1 snRNP bound at the 5' splice site, the U2AF 35/65 heterodimer bound at the polypyrimidine tract and 3' splice site, and the U2 snRNP bound to the branch point. This A complex

recruits the U4/U5/U6 tri-snRNP and other components of the full B complex spliceosome, before undergoing extensive rearrangements and compositional changes to form the catalytic spliceosome (called B$^{act}$) with the intron splice sites assembled into the catalytic center of the structure (*Will and Lührmann, 2011*). In contrast to this pathway of intron assembly, a typical transcript in animal cells has very long introns, and the initial recognition of splice sites and prespliceosome assembly occur on the short exons to form exon definition complexes (EDCs) (*Berget, 1995*; *De Conti and Baralle, 2012*). These EDCs then interact across a long intron to form the catalytic spliceosome. The EDC contains the U1 snRNP, the U2 Auxiliary Factor (U2AF), and the U2 snRNP bound to the 5' splice site, the 3' splice site, and the branch point, respectively (*Berget, 1995*; *De Conti et al., 2012*; *Robberson et al., 1990*; *Chiara and Reed, 1995*; *Hoffman and Grabowski, 1992*). The EDC is sometimes called an A-like complex or an exon-defined A complex (EDA) because of its similar size and composition to an intronic prespliceosome (*Berget, 1995*; *Sharma et al., 2008*; *Schneider et al., 2010*; *House and Lynch, 2006*). Like an intronic prespliceosome, the binding of the U2 snRNP to the branch point in the EDC is preceded by U1 snRNP and U2AF binding and requires ATP (*Will and Lührmann, 2011*; *Berget, 1995*; *De Conti et al., 2012*; *Staknis and Reed, 1994*; *Michaud and Reed, 1993*). Early steps in assembly of the EDC are poorly characterized. For example, the roles and fates of the hnRNP and SR proteins, which bind before EDC completion, are unknown. Elucidating the sequence of events leading to this intricate ribonucleoprotein (RNP) is a prerequisite to understanding the mechanisms of splicing regulation.

One extensively studied splicing regulator is the polypyrimidine-tract binding protein 1 (PTBP1, also called PTB or hnRNP I). PTBP1 and its paralogs PTBP2 and PTBP3 control a variety of developmental splicing programs and have been particularly studied in neuronal, muscle, and testis development (*Zheng et al., 2012*; *Linares et al., 2015*; *Keppetipola et al., 2012*; *Bland et al., 2010*; *Hall et al., 2013*; *Boutz et al., 2007*; *Li et al., 2014*; *Spellman et al., 2005*; *Zagore et al., 2015*). PTBP1 contains four RRM-type RNA binding domains, each binding a triplet of C and U nucleotides, giving the full protein a high affinity for runs of mixed pyrimidines (*Oberstrass et al., 2005*; *Auweter and Allain, 2008*). Exons repressed by PTBP1 usually have multiple binding sites found in a variety of locations relative to the target exon (*Linares et al., 2015*; *Xue et al., 2009*; *Llorian et al., 2010*; *Han et al., 2014*; *Amir-Ahmady et al., 2005*). The mechanism of PTBP1-induced exon skipping seems to vary with the location of the binding sites (*Keppetipola et al., 2012*; *Spellman and Smith, 2006*; *Wagner and Garcia-Blanco, 2001*). A PTBP1-binding site within exon 6 of Fas pre-mRNA blocks the U1 snRNP-dependent recruitment of U2AF65 to the upstream 3' splice site (*Izquierdo et al., 2005*). In other exons, including exon 6B of β-tropomyosin pre-mRNA and exon 10 of GABA$_A$ receptor γ2 pre-mRNA, PTBP1 directly competes with U2AF65 for binding to the polypyrimidine tract of the 3' splice site (*Saulière et al., 2006*; *Ashiya and Grabowski, 1997*; *Singh et al., 1995*). Repression of the Src N1 exon requires two binding sites, one embedded in the polypyrimidine tract of the 3' splice site and one in the downstream intron (*Amir-Ahmady et al., 2005*; *Chan and Black, 1997*; *Chou et al., 2000*; *Chan and Black, 1995*). These sites each bind a least one PTBP1 monomer. The upstream site binds with high affinity and appears to block U2AF binding similar to what is seen with β-tropomyosin exon 6B, but is not sufficient for full splicing repression (*Amir-Ahmady et al., 2005*; *Chan and Black, 1995*). The required downstream intronic site binds PTBP1 with lower affinity but is stimulated for binding by the upstream site, implying an interaction between the PTBP1 proteins bound at the two sites (*Amir-Ahmady et al., 2005*). An exon repressed by these two sites still recruits the U1 snRNP to its 5' splice site, but this U1 snRNP fails to interact with and splice to an EDC on exon 4 downstream (*Sharma et al., 2008*). Interestingly, the downstream PTBP1 directly contacts stem-loop 4 of the U1 snRNA bound to the N1 exon, and this interaction is proposed to block the interaction with the exon 4 complex further downstream (*Sharma et al., 2011*, *2014*).

Unlike the exons described above, many PTBP1-repressed exons have binding sites at relatively distal positions within each flanking intron that would not interfere with the target exon or its splice sites (*Linares et al., 2015*; *Xue et al., 2009*; *Llorian et al., 2010*; *Han et al., 2014*; *Spellman and Smith, 2006*). In one model, the simultaneous binding to these sites is proposed to create an RNA loop that somehow excludes the exon from the splicing apparatus (*Oberstrass et al., 2005*; *Spellman and Smith, 2006*; *Chou et al., 2000*). Alternatively, PTBP1 was proposed to multimerize across the exon between the two binding sites to prevent exon complex assembly (*Spellman and*

*Smith, 2006*; *Wagner and Garcia-Blanco, 2001*). However, this has not been tested by direct examination of exon RNP complexes and of how their assembly is altered by the splicing regulator.

In earlier work, we found that Src alternative exon N1 failed to properly interact with the active EDC assembled onto the constitutive exon 4 downstream, despite the presence of the U1 snRNP bound to the 5′ splice site of the N1 exon (*Sharma et al., 2008*). In that study, we could not dissect changes in RNP complexes assembled on the N1 exon that were caused by PTBP1 because of the presence of exon 4 in *cis*. To examine individual exon complexes, we developed an in vitro *trans*-splicing assay for a model PTBP1-repressed exon that has distal flanking PTBP1-binding sites in both introns. We found that PTBP1 binding allows normal U2AF65 and U1 snRNP recruitment to this exon, but prevents U2 snRNP assembly and EDC formation. Using SILAC-MS, we identified extensive compositional differences between a PTBP1-repressed exon complex, an active early exon complex, and an EDC with bound U2 snRNP. PTBP1 acts to prevent the remodeling of an early exon RNP during exon definition.

## Results

### Distal PTBP1 binding upstream and downstream of an alternative exon represses both splice sites

In earlier work, we found that PTBP1-binding sites upstream and downstream of an otherwise efficiently spliced exon would repress its splicing (*Amir-Ahmady et al., 2005*). Both PTBP1 sites were required for splicing repression, and both were active even when separated from the splice sites of the exon, indicating that PTBP1 did not simply block binding of spliceosomal components. Genome-wide binding analyses by crosslinking immunoprecipitation (CLIP) identify many such PTBP1-dependent exons with distal flanking binding sites (*Xue et al., 2009*; *Llorian et al., 2010*; *Han et al., 2014*). In one study, 95 exons were identified in mouse ES cells that were both derepressed upon PTBP1 depletion and bound PTBP1 in iCLIP analysis (*Supplementary file 1*) (*Linares et al., 2015*). Of these exons, 56 (59%) exhibited PTBP1 binding only in distal regions of the flanking introns, and most of these exhibited binding both upstream and downstream. No exons were seen to bind PTBP1 only in the exon itself or its 3′ splice site.

To examine the splicing and RNP assembly of an exon with distal PTBP-binding sites, we modified the alternative exon N1 of Src, which has adjacent PTBP1-binding sites that repress its splicing in non-neuronal cells (*Chan and Black, 1997*; *1995* ; *Chou et al., 2000*). We shifted the position of the PTBP1-binding site upstream of N1 away from the polypyrimidine tract of the 3′ splice site to a position 25-nt upstream of the branch point (*Figure 1A*) that should not hinder U2 snRNP binding (*Gozani et al., 1996*). We replaced the polypyrimidine tract that contained this PTBP1 site with the 3′ splice site of the Adenovirus Major Late first intron to allow efficient U2AF65 recruitment without competition with PTBP1. We previously showed that upstream PTBP1 could repress this splice site in an in vivo reporter transcript (*Amir-Ahmady et al., 2005*). The 5′ splice site at the downstream side of N1 was also modified to match the consensus splice site, which splices more efficiently than the native N1 site, and to remove potential cryptic sites nearby (*Chou et al., 2000*). The sequence downstream from N1, which contains PTBP1-binding sites required for splicing repression, was left intact. The strong splice sites in the test exon allow for more efficient splicing in trans-splicing assays (see below), and we previously found that similar constructs containing these sites were indeed repressible by PTBP1 (*Amir-Ahmady et al., 2005*; *Chou et al., 2000*). PTBP1-mediated splicing repression of this test exon was confirmed in *in vitro* splicing assays using a two-exon substrate containing the N1 exon and the downstream constitutive Src exon 4, with either a wildtype or mutant PTBP1-binding site upstream. Mutation of the PTBP1-binding site enhanced splicing of the substrate (*Figure 1— figure supplement 1* and see below).

To examine the effect of PTBP1 specifically on the test exon, it was transcribed with adjacent intron sequences as an independent transcript and assayed for reactivity in a *trans*-splicing reaction. This assay allows measurement of splicing activity of both splice sites of the test exon. It may also provide a better model of splicing within a multi-exon vertebrate pre-mRNA, where long introns separate short exons that must undergo exon definition (*Berget, 1995*). We incubated the N1 exon transcripts containing either the wildtype or mutant upstream PTBP1-binding site in HeLa nuclear extract to pre-assemble exon RNP complexes (*Figure 1B and C*). In parallel reactions, transcripts

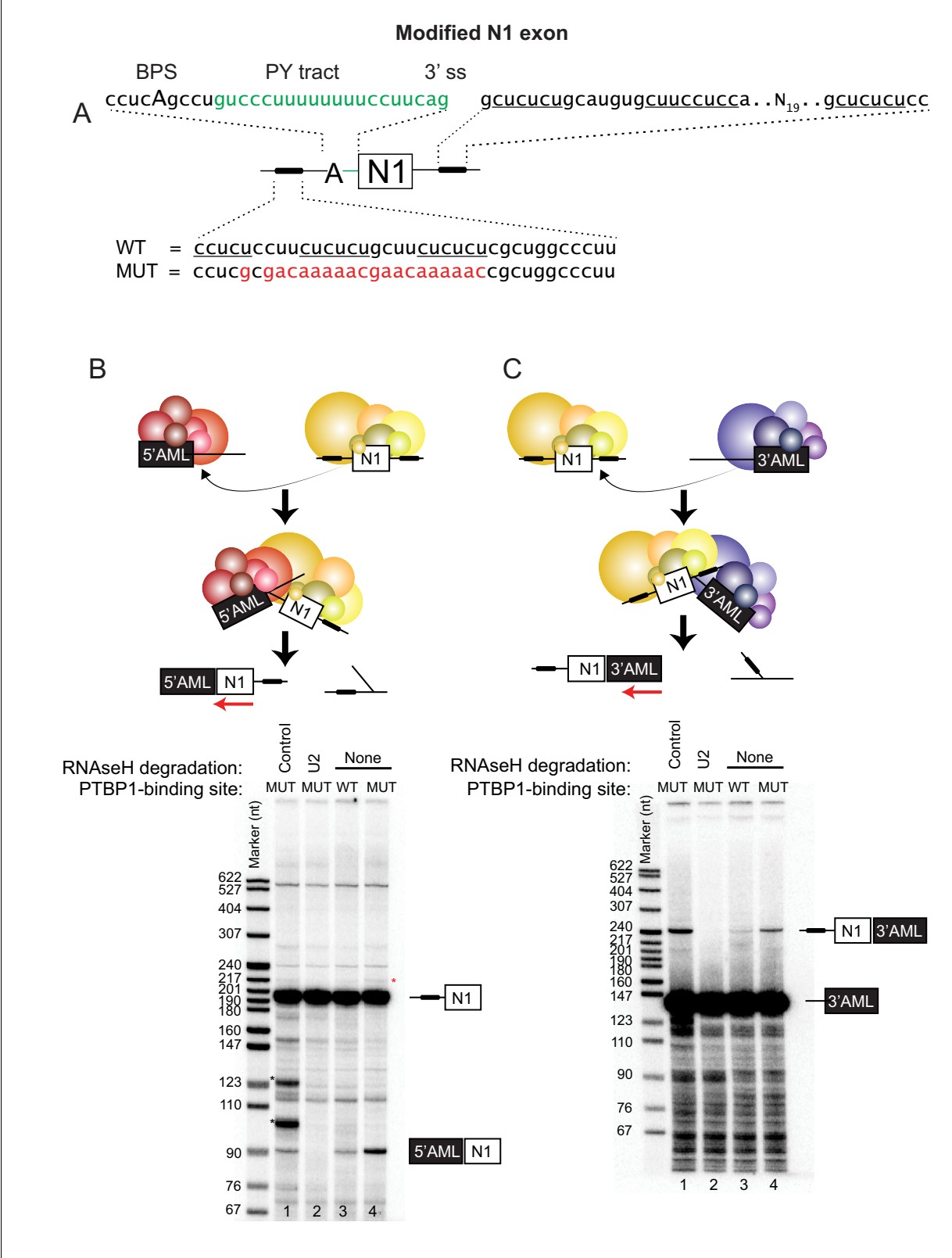

**Figure 1.** Distal PTBP1 binding flanking an alternative exon represses both splice sites. (**A**) Diagram of the modified Src N1 exon RNA used in this study. PTBP1-binding sites are indicated as black bars. The upstream site (shown below) originally located within the polypyrimidine tract of the 3' splice site was shifted to a position 25-nt upstream of the branch point sequence (BPS *[Black, 1991]*; cucAg). The polypyrimidine tract (PY tract) and 3' splice site (3' ss) of the Adenovirus Major Late (AML) first intron (highlighted in green) were added in place of the original N1 splice site. The
*Figure 1 continued on next page*

*Figure 1 continued*

downstream PTBP1-binding sites were left unchanged (shown above to the right). A mutation (MUT) of the upstream PTBP1-binding sites known to prevent PTBP1 binding is highlighted in red (*Sharma et al., 2008*; *Amir-Ahmady et al., 2005*). Nucleotides known to be important for PTBP1 binding are underlined. (B) Top panel: Diagram of the *trans*-splicing assay between the N1 exon and the 5′ splice site of the first AML intron (5′AML). Complexes assembled on either the WT or MUT N1 exon were mixed with the 5′AML complex and further incubated. Total RNA was extracted and assayed by primer extension analysis with a $^{32}$P labeled 3′ exon primer (red arrow). Bottom panel: primer extension products were resolved on 8% urea-PAGE. Spliced (90 nts.) and unspliced (195 nts) products are indicated. Bands marked with a black asterisk (*) resulted from an unexpected RNAse H cleavage of the N1 RNA by the control oligo, while a red asterisk (*) indicates self *trans*-splicing of the N1 exons. (C) Top panel: Diagram of the *trans*-splicing assay between the N1 exon and the downstream 3′ splice site of the first AML intron (3′AML). Reaction conditions were as in (B). Bottom panel: urea-PAGE of the primer extension assay as in (B), except using an oligo complementary to the 3′AML exon. Spliced (246 nts.) and unspliced (140 nts.) products are indicated.

The following figure supplement is available for figure 1:

**Figure supplement 1.** Distal PTBP1 binding represses splicing of the N1 exon.

containing portions of the first or the second exon of the Adenovirus Major Late pre-mRNA (AML), including either the 5′ splice site or the 3′ splice site, were also pre-incubated in the extract. To enhance splicing of these Adeno transcripts relative to the splicing of N1 to itself, the Adeno RNAs (50 nM) were in 25-fold excess over the N1 exon RNA (2 nM). This is lower than the RNA concentration used for many *trans*-splicing reactions and lower than the approximate concentration of the U1 snRNP (200 nM) (*Chiara and Reed, 1995*; *Chen et al., 2000*; *Konarska and Sharp, 1987*). It is still possible that the excess Adeno RNA also affects splicing by titrating RNA binding proteins, but this effect should be the same for reactions containing either wild-type and mutant N1 exons. After pre-incubation the two sets of reactions were mixed, and incubation continued under standard splicing conditions to allow *trans*-splicing between the N1 exon and either the Adeno 5′ or 3′ splice site. After RNA isolation, spliced products were analyzed by primer extension using primers annealed to the 3′ exon of the products.

The N1 RNA lacking the upstream PTBP1-binding site was seen to splice to either of the Adeno splice sites (*Figure 1B* lane 1 at 90 nts. and 1C lane 1 at 246 nts). That these products resulted from splicing was confirmed by RNase H-mediated depletion of the U2 snRNP, which led to loss of these bands (*Figure 1B and C* cf. lane 1 to lane 2). The formation of the *trans*-spliced product between the N1 exon and the 5′ AML exon was strongly inhibited by the upstream PTBP1 site (*Figure 1B* cf. lane 3 to 4). Notably, the upstream PTBP1 site also inhibited *trans*-splicing between the downstream N1 exon 5′ splice site and the 3′ AML exon (*Figure 1C* cf. lane 3 to 4). The inhibition of the N1 exon 5′ splice site by the upstream PTBP1 site was observed previously in *cis*-splicing reactions (*Chan and Black, 1997*; *Chou et al., 2000*), also indicating that the protein is not simply blocking the splice sites. A *trans*-splicing reaction of the N1 exon to itself, which produced a 218-nt spliced product when probed with an N1-specific primer, could be detected only very weakly and was also inhibited by the upstream PTBP1-binding site (*Figure 2B* red asterisk). From these *trans*-splicing assays, we conclude that both splice sites of the N1 exon construct are active for splicing, but are inhibited by PTBP1.

## PTBP1 inhibits assembly of the exon definition complex

To examine how protein assembly changes as a result of PTBP1 repression, we used native gel electrophoresis to analyze exon RNP complexes containing the N1 exon RNA in the PTBP1-repressed and active states. The wildtype transcript bound by PTBP1 formed a typical heterogeneous complex accompanied by a small amount of a more slowly migrating complex (*Figure 2A* lane 1–4). In contrast, the mutant transcript that lacked the upstream PTBP1-binding site and was active for splicing formed much more of the larger exon complex. The mutant transcript initially formed the faster migrating complexes that were converted to the larger complex over time (*Figure 2A* lane 5–8). This larger complex was ATP-dependent and required intact U1 and U2 snRNPs, confirming its identity as an exon definition complex (EDC) (*Figure 2B* cf. lane 2 to lane 4 and 6; *Figure 2—figure supplement 1*). To further confirm that the lack of EDC formation on the wildtype transcript was due to PTBP1 binding, we immuno-depleted PTBP1 from the nuclear extract. Depletion of PTBP1

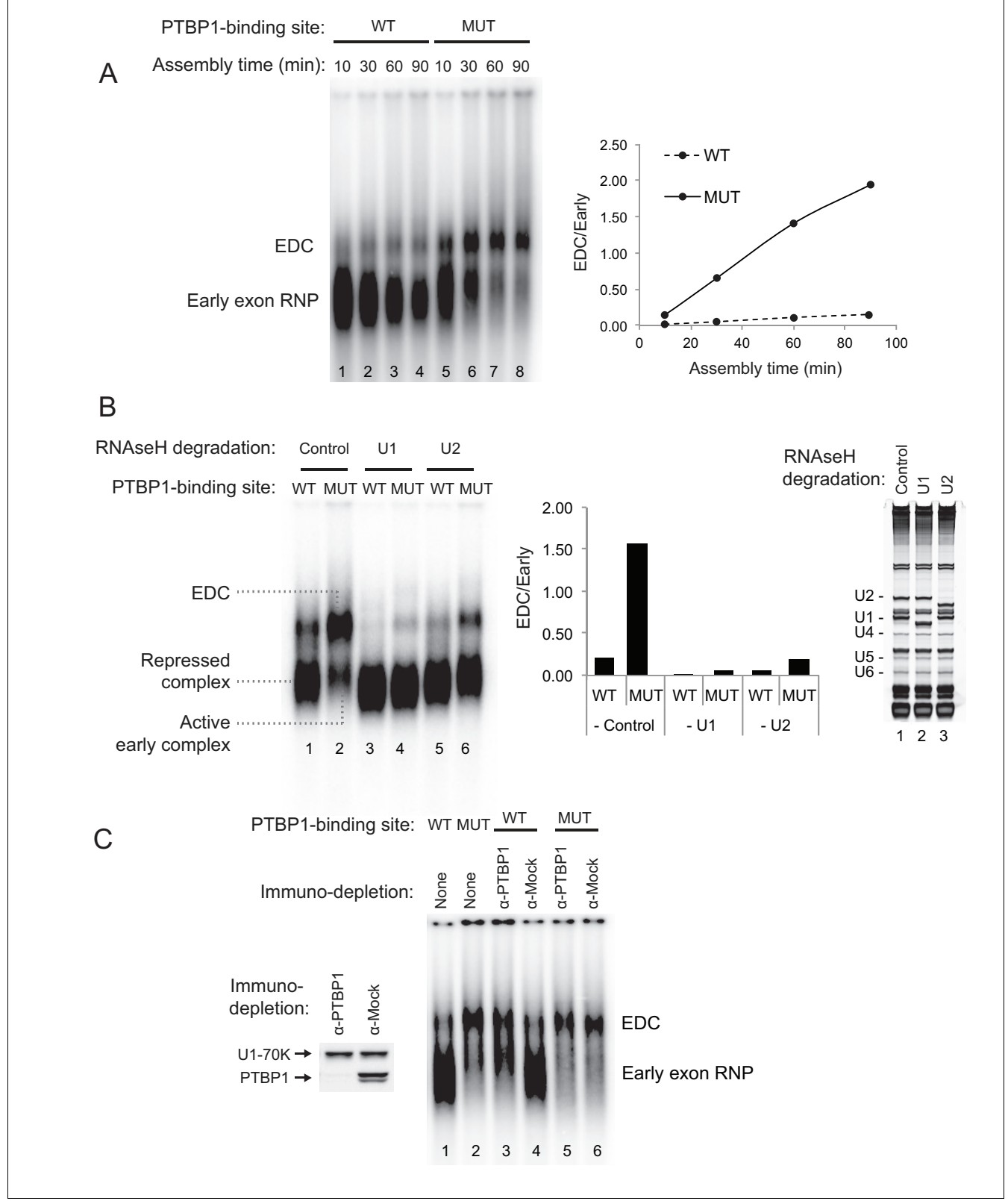

**Figure 2.** PTBP1 inhibits assembly of the exon definition complex. (**A**) Left panel: Native gel assay of N1 exon RNAs containing WT or MUT PTBP1-binding sites. RNAs were incubated in nuclear extract under splicing conditions for the indicated times to assemble exon RNP complexes. The early exon RNP and the exon definition complex (EDC) are labeled. Right panel: The ratio of the EDC to early exon RNP was plotted over time. (**B**) The EDC complex is dependent on the U1 and U2 snRNPs. WT and MUT N1 exon RNAs were incubated in nuclear extracts pre-treated with RNAse H and DNA

*Figure 2 continued on next page*

*Figure 2 continued*

oligos complementary either to the U1 (nt. 1–15), U2 (nt. 1–15) snRNAs or GAPDH mRNA (control). Right panel: SYBR Gold-stained urea-PAGE of total RNA in the extract shows the efficiency of snRNA degradation. Left panel: Native gel assay of the WT and MUT N1 exons assembled in different pre-treated extracts for 30 min. The early exon RNP from WT and MUT RNAs are indicated. Middle panel: the ratio of EDC to early exon RNP was plotted for each reaction to the left. (C) Immuno-depletion of PTBP1 allows EDC assembly. Prior to complex assembly, nuclear extracts were either mock-depleted (α-Mock) or PTBP1-depleted (α-PTBP1). An immunoblot on the left shows efficiency of PTBP1 depletion. U1-70K was used as a loading control. The pre-treated extracts were then used for assembling exon complexes as in (B).

The following figure supplement is available for figure 2:

**Figure supplement 1.** The exon definition complex is ATP-dependent.

stimulated the assembly of the EDC on the wildtype exon, whereas mock-depleted extract remained inhibited (*Figure 2C* cf. lane 3 to 4). PTBP1 depletion or mock depletion did not affect formation of the EDC on the mutant substrate (*Figure 2C* cf. lane 5 to 6).

These analyses defined exon complexes in three regulatory states. 1) A repressed complex that assembles early onto the wildtype RNA in the presence of PTBP1 and does not progress further. 2) An active early complex that assembles onto the mutant exon lacking upstream PTBP1, and then converted into 3) the exon definition complex. The gel system did not resolve differences between the repressed complex and the active early complex. The reactions were treated with heparin prior to gel separation to reduce aggregation and help material enter the gel. Heparin can strip U1 and other factors from the RNA and prevent resolution of complexes similar to the 'E complex', previously defined for spliceosome assembly onto introns (*Reed, 1990*; *Michaud and Reed, 1991*; *Michaud and Reed, 1993*; *Sharma et al., 2005*; *Das and Reed, 1999*; *Kent and MacMillan, 2002*). We expect the U1 snRNP to be in both complexes, as we previously found that U1 bound to the N1 exon even when it was repressed by PTBP1 (*Sharma et al., 2005*). To examine their components, the three N1 exon RNP complexes were analyzed further.

## PTBP1 blocks assembly of the U2 snRNP but not of U2AF65 or the U1 snRNP

Previously described exon definition complexes contained the U1 and U2 snRNPs base-paired to the 5' splice site and the branch point, respectively. They also contained the U2AF heterodimer (U2AF65 and U2AF35) bound to the polypyrimidine tract and AG di-nucleotide of the 3' splice site, as well as a large number of additional protein factors whose roles are less defined (*Sharma et al., 2008*; *Schneider et al., 2010*). To examine the composition of the three N1 exon complexes, we tagged the mutant and wildtype RNAs with the MS2 stem-loop to allow their purification (*Figure 3A*). As described previously, the RNAs were pre-bound with the MS2 coat protein-maltose binding protein fusion (MS2-MBP), and then assembled into exon RNP complexes under standard splicing conditions (*Sharma et al., 2008*; *Jurica et al., 2002*; *Das R, 2000*). Reactions were incubated for 30 min to allow isolation of both the active early and EDC complexes. Complexes were resolved by ultracentrifugation in glycerol density gradients. As seen on native gels, the RNAs generated the two complexes as separable peaks on the gradient (*Figure 3A*). A larger percentage of the mutant RNA was converted to the EDC (and its aggregates at the bottom) compared to the wildtype RNA. Peak fractions for each complex were pooled and affinity purified on amylose resin. RNA from each complex was analyzed on urea-PAGE stained with SYBR Gold. As expected, the repressed and active early complexes contained only the U1 snRNA and the exon RNA substrate, and these were present in equimolar amounts (*Figure 3B* lane 1, 2 and graph). This is in agreement with results showing that the U1 snRNP is bound to the repressed N1 exon, where it interacts with PTBP1 (*Sharma et al., 2011*). In contrast, the purified EDC contained both the U1 and U2 snRNAs in a precise 1:1 stoichiometry relative to the N1 exon RNA (*Figure 3B* lane 3 and graph). This is similar to the prespliceosomal A complex assembled onto an intron, but the splice sites flank an exon and thus the EDC has an inverted 5'-3' orientation compared to the A complex. Similar to our previous observations with an EDC from Src exon 4, snRNAs of the U4/5/6 tri-snRNP were not observed in the (*Sharma et al., 2008*). The U4/5/6 tri-snRNP was previously found in an EDC derived from an Adenovirus exon

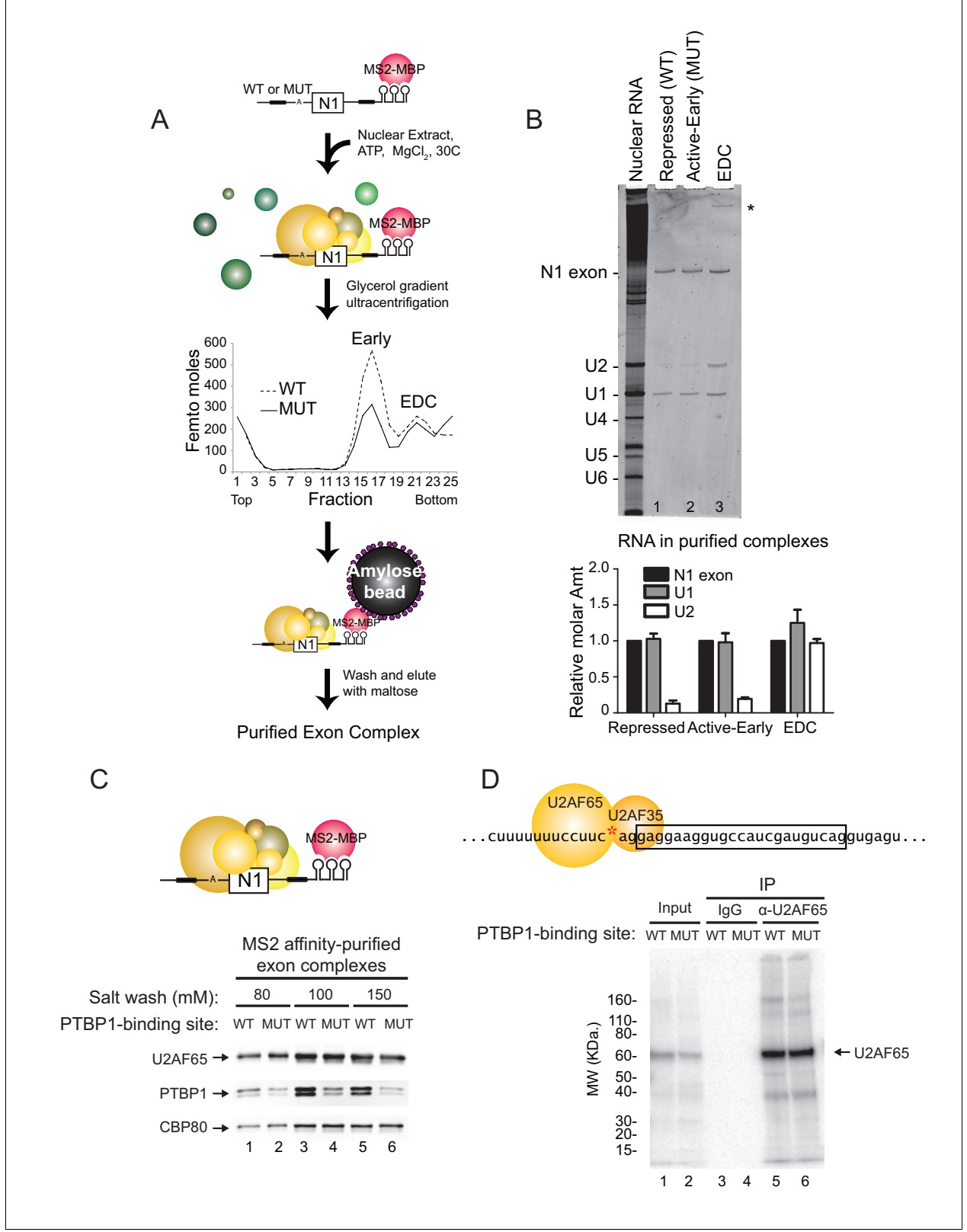

**Figure 3.** PTBP1 blocks assembly of the U2 snRNP but not U2AF65 or the U1 snRNP. (**A**) Diagram of exon complex purification. MS2-tagged WT and MUT N1 exons were incubated with the MS2-MBP fusion protein prior to assembling of exon complexes. The reactions were resolved on glycerol density gradients, peak fractions were pooled, and complexes were affinity purified on amylose beads. Note that the MUT RNA also forms larger complexes found at the bottom of the gradient. (**B**) Top panel: RNA was extracted from exon complexes purified as in (**A**), resolved on 8% urea-PAGE,

*Figure 3 continued on next page*

*Figure 3 continued*

and stained with SYBR Gold. Total nuclear RNA is shown to the left with each U snRNA indicated. The asterisk (*) marks uncharacterized band. Bottom panel: Intensities of the N1 exon, U1 and U2 bands normalized by their nucleotide length are plotted relative to the N1 exon for each complex. (**C**) MS2-tagged WT and MUT N1 exon RNAs were assembled into exon complexes in the absence of ATP to prevent progression to the EDC, purified on amylose beads, washed in increasing concentrations of potassium glutamate (80, 100 or 150 mM) and eluted in maltose. Proteins in each complex were resolved on SDS-PAGE and immunoblotted with antibodies against PTBP1, U2AF65 and CBP80 (loading control). (**D**) Diagram of the 3' splice site, labeled at the AG (red asterisk). Labeled WT and MUT RNAs were assembled in splicing reactions for 10 min before irradiation in 254-nm UV to crosslink proteins to the RNAs. Reactions were treated with RNAse T1, denatured in 0.1% w/v SDS at 95°C, immuno-precipitated with α-U2AF65 antibody or mouse IgG (control), resolved on SDS-PAGE, and autoradiographed. The band of crosslinked U2AF65 is indicated.

The following figure supplement is available for figure 3:

**Figure supplement 1.** Recruitment of U2AF65 is U1-dependent regardless of the presence of PTBP1.

---

(*Schneider et al., 2010*). It is possible that the tri-snRNP binds to the N1 exon in a later step, or that the N1 EDC and the Adeno EDC differ in their composition.

The recruitment of the U2 snRNP requires the prior binding of the U2AF heterodimer to the 3' splice site. Previous studies of an exonic PTBP1-binding site found that the exon-bound PTBP1 prevented the U1 snRNP-dependent recruitment of U2AF65 to the upstream polypyrimidine tract (*Izquierdo et al., 2005*). Similarly, PTBP1 bound within the polypyrimidine tract of the 3' splice site itself is thought to directly block U2AF binding. To examine this in an exon with distal PTBP1 binding sites, we measured the levels of U2AF65 within the MS2 affinity-purified exon complexes by immunoblot (*Figure 3C*). These were normalized to CBP80, which binds in equimolar amounts to the 5' cap of each RNA. Compared to PTBP1, which is expected to bind at higher levels to the wildtype RNA than the mutant RNA missing one of the PTBP1 binding sites, U2AF65 bound to the RNAs in equal amounts regardless of PTBP1 binding (*Figure 3C* cf. lane 1 to 2, 3 to 4, 5 to 6). This U2AF65 binding was not affected by moderate changes in salt concentration during isolation (80 to 150 mM). Similar levels of U2AF65 bound to wildtype and mutant RNA were also seen in mass spec data of the repressed and active early complexes (see below).

To confirm that U2AF was binding correctly to the N1 exon 3' splice site and not a cryptic site, we used site specific labeling to assay U2AF65 crosslinking to the N1 3' splice site. The wildtype and mutant transcripts were specifically $^{32}$P-labeled 5' of the A nucleotide within the 3' splice site AG (*Figure 3D* red asterisk). The RNAs were assembled into exon complexes under standard splicing conditions and then UV-irradiated to crosslink proteins to RNA. Crosslinked products were digested with RNAse T1 and resolved on SDS-PAGE. For both the wildtype and mutant transcripts, the prominent crosslinked protein migrated slightly above 60 KDa (*Figure 3D* lane 1 and 2). Immuno-precipitation with anti-U2AF65 antibody confirmed that this major crosslinked product was of U2AF65 and that the protein crosslinked equally to the 3' splice site of the wildtype and mutant transcripts (*Figure 3D* cf. lane 5 to 6).

Previous studies found that the recruitment of U2AF to a 3' splice site was dependent on binding of the U1 snRNP to the 5' splice site downstream (*Izquierdo et al., 2005*; *Misra et al., 2015*). To examine this, we isolated N1 exon complexes after the depletion of U1 snRNA with RNAse H (*Figure 3—figure supplement 1*). As seen previously, the recruitment of U2AF65 to the polypyrimidine tract of both the wildtype and the mutant exon was U1-dependent (*Figure 3—figure supplement 1* cf. lane 1 to 2, and lane 3 to 4). However, this recruitment was not prevented by PTBP1 binding (cf. lane 1 to 3).

These data indicate that distal PTBP1 binding allows normal U1 snRNP binding and U2AF recruitment onto the repressed exon, but further assembly to an EDC is blocked at a step prior to U2 snRNP binding.

## Distal PTBP1 binding causes dramatic compositional changes in exon complexes

The early exon RNP complexes assembled onto the wildtype and mutant exon RNAs have equal amounts of bound U1 and U2AF, and appear similar in gel mobility, but differ in their ability to further assemble into the EDC. To quantitatively compare their composition, we used SILAC-mass

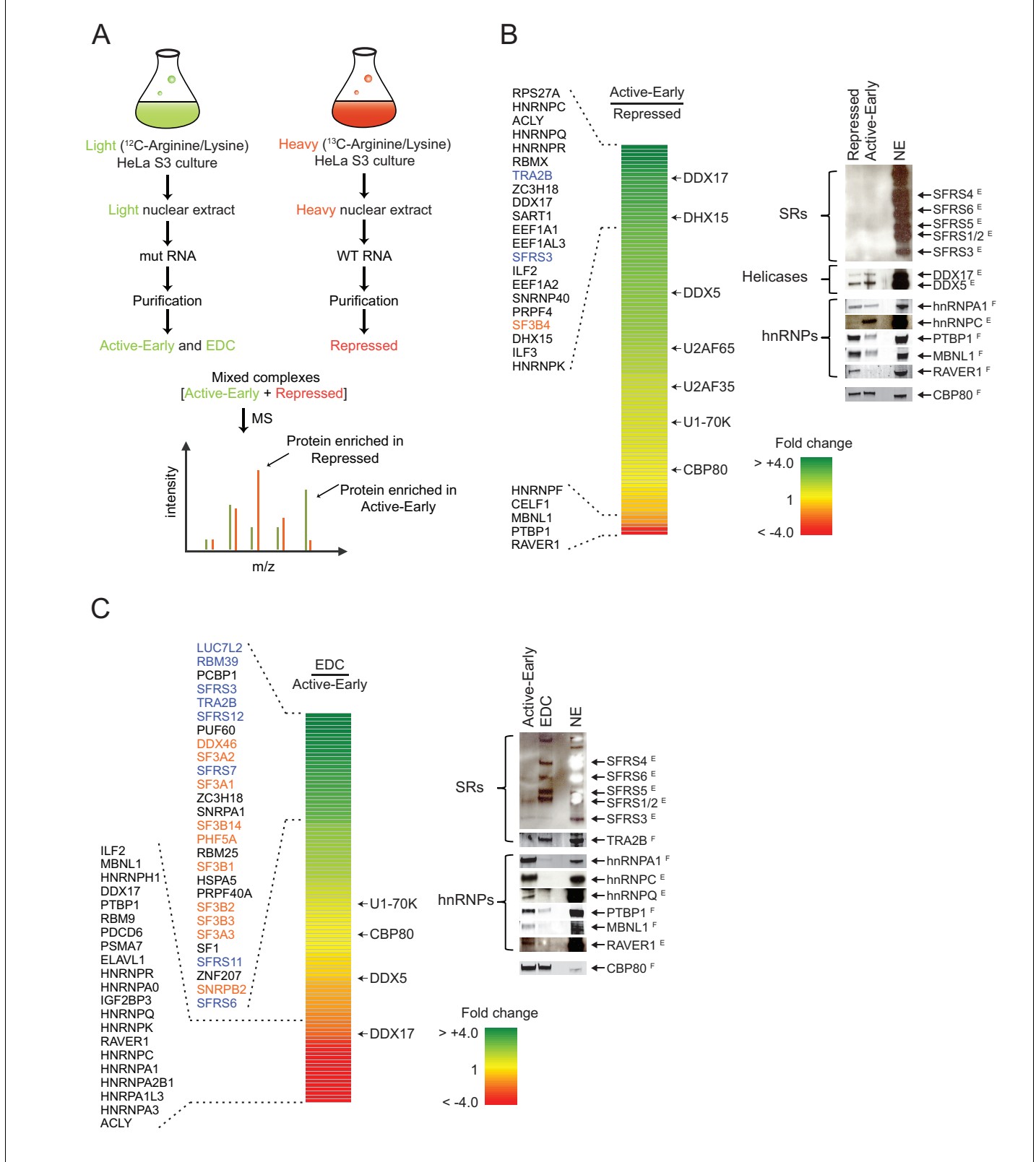

**Figure 4.** Distal PTBP1 binding causes dramatic compositional changes in exon complexes. (**A**) Diagram of the SILAC-MS experiment. MS2-tagged WT and MUT N1 exons were used for assembling exon complexes with heavy (red) and light (green) nuclear extracts, respectively. The heavy repressed and light active early complexes were purified as in **Figure 3A**, and then mixed in a 1:1 molar ratio and subjected to MS analysis. (**B**) The relative abundance of proteins in the active early over the repressed complex. Left panel: A heat map showing the fold changes of individual proteins found in both

*Figure 4 continued on next page*

*Figure 4 continued*

complexes. Fold changes were calculated by dividing a normalized SILAC ratio from the active complex by the SILAC ratio from repressed complex. A color scale for the fold-change scale is shown. Proteins enriched in the active early complex are in green; those enriched in the repressed complex are in red, and proteins equally present in the two complexes are in yellow. Proteins with substantial fold changes are labeled on the left (blue: SR proteins; orange: U2 snRNP proteins). Right panel: Immunoblot of proteins of interest in the repressed and active early complexes. CBP80 was used as a loading control. Superscripted 'E' indicates ECL (enhanced chemiluminescence) detection, while 'F' indicates fluorescent secondary antibodies. Note that changes in band intensity seen by ECL are less quantitative than for fluorescence. ECL was used to increase the signal for antibodies that were less effective. (C) Relative abundance of proteins in the EDC over the active early complex. Left panel: A heat map showing the fold change of individual proteins found in both complexes. A color scale is below. Proteins are labeled on the left as in (B). Right panel: Immunoblot of proteins of interest in the EDC and active early complexes. NSAF values for all proteins are given in *Supplementary file 1*.

spectrometry (stable isotope labeling by amino acids in cell culture). HeLa S3 cells were cultured in either standard $^{12}$C amino acid media or $^{13}$C-Arginine and $^{13}$C-Lysine-substituted media (*Figure 4A*). Nuclear extracts were prepared from these light ($^{12}$C) or heavy ($^{13}$C) cultures and used for assembling exon complexes. The wildtype RNA was incubated in the heavy nuclear extract to form the repressed complex, and the mutant RNA was incubated in the light extract to form to the active early complex and the EDC. Exon RNP complexes were purified as above, and the heavy repressed complex and the light active early complex were then mixed in a 1:1 molar ratio based on the amount of exon RNA. The mixed complexes were subjected to trypsin digestion and LC-MS-MS analysis to comprehensively identify their components. Each trypsinized peptide was detected as two peaks derived from $^{12}$C or $^{13}$C substituted Lysine or Arginine (*Figure 4A*). The SILAC ratio (light over heavy) was calculated from the intensities of these two peaks to provide a measure of the relative representation each protein in the two complexes. Averaged SILAC ratios of Cap binding complex proteins, CBP20 and CBP80 that bind equally to all the exon RNP complexes were used to normalize the SILAC ratios of other proteins and remove errors resulting from unequal mixing and sample processing of the two complexes.

Many proteins had SILAC ratios near 1 indicating equal binding in the repressed and active early complexes. As expected, these included U1-specific proteins, U1A, U1C and U1-70K. U2AF65 and U2AF35 exhibited slightly higher levels in the active early exon complex, possibly due to the stabilization of their binding by other factors as this exon assembles further into a complex active for splicing (*Figure 4B*). Of the proteins exhibiting large differences in binding between the two complexes, only three showed a greater than two-fold enrichment in the repressed complex. These were the expected PTBP1 and two proteins known to be PTBP1 corepressors in the repression of other exons, RAVER1 and MBNL1 (*Figure 4B* right panel; *Supplementary file 1*) (*Joshi et al., 2011*; *Gromak et al., 2003*; *Gooding et al., 2013*). Another known PTBP1 interacting protein, MATRIN3, was inconsistently found in the exon complexes (*Coelho et al., 2015*).

Many proteins exhibited the opposite pattern of binding from PTBP1, MBNL1, and RAVER1, with increased binding in the active early complex compared to the repressed complex. Large increases were seen in multiple hnRNP proteins including C, D, G (RBMX), K, R, Q (SYNCRIP) and U (*Figure 4B*; *Supplementary file 1*). HnRNP A1 and H, previously shown to bind the N1 exon and downstream, did not change substantially between the two complexes (*Rooke et al., 2003*; *Markovtsov et al., 2000*; *Chou et al., 1999*). Interestingly, SR proteins were present at only low levels in the two early complexes as measured by normalized spectral abundance factor (NSAF) and by immunoblot (*Figure 4B* right panel; *Supplementary file 1*). Although not abundant, SFRS3 and TRA2B (SFRS10) did increase in the active early complex. The presence of proteins in the repressed and active early complexes was confirmed by immunoblot and additional mass spec analyses (*Figure 4B* right panel and see below).

Several ATP-dependent RNA helicases were present in the complexes. Some of these are known to function in later steps of spliceosome assembly. Of particular interest was DDX17 (p72), which increased 3-fold in the active complex along with a smaller increase in its paralog DDX5 (p68) (*Figure 4B*; *Supplementary file 1*). These proteins are known to be required for the splicing of several alternative exons, but the targets of their activity are not well defined (*Kar et al., 2011*; *Dardenne et al., 2012*; *Dardenne et al., 2014*). They were previously identified in other exon complexes and co-purified with the U1 and U2 snRNPs (*Sharma et al., 2008*; *Schneider et al., 2010*;

*Kar et al., 2011*; *Jurica and Moore, 2003*; *Hartmuth et al., 2002*; *Chu et al., 2015*). Their recruitment into the active early exon complex prior to EDC formation may indicate a role in the remodeling we observed in this complex.

To further validate the compositional differences between the exon complexes and their PTBP1 dependence, we performed a second MS experiment. In this experiment, we assembled complexes only on the wildtype RNA in light HeLa nuclear extract that had been either immuno-depleted for PTBP1 or mock-depleted. This produced the active early complex and the EDC in the PTBP1-depleted extract and the repressed complex in the mock-depleted extract (*Figure 2C* lane 3 and 4). Prior to MS analysis, these three light complexes were spiked with a mixture of exon RNP complexes MS2-affinity purified from heavy extract containing both the wildtype and mutant RNAs. This mixture generated a heavy peptide peak for all the proteins of interest. Peak intensities for the three light complexes were then normalized to peak intensities of the heavy CBP20 and CBP80 to measure the relative abundance of each protein in each light complex.

Many proteins, including those described above, showed consistent changes between the two MS experiments (*Supplementary file 1*). In the first experiment, the repressed and active early complexes contained the wildtype RNA and the PTBP1-binding-site mutant RNA, respectively. In the second experiment, the proteins were assembled onto the wildtype RNA plus or minus PTBP1. Thus, proteins seen changing in both experiments are responding to the differential binding of PTBP1. Proteins observed in only one mass spectrometry experiment were not considered further. These results indicate that despite their similar gel mobility, the PTBP1-repressed complex and the active early complex, which can progress to the EDC, are very different. The binding of PTBP1 in conjunction with MBNL1 and RAVER1 prevented the binding of many other proteins, interestingly including many hnRNP proteins. The binding of U2 snRNP to form the EDC is preceded by extensive remodeling of the exon RNP from the repressed to active state.

We next examined the compositional changes during the transition of the active early exon complex to the EDC. We mixed the heavy repressed complex and the light EDC from the first MS experiment in a 1:1 molar ratio and carried out the trypsin digestion and LC-MS-MS. The SILAC ratio of each protein (light over heavy) was calculated from peptide peak intensities again normalized to CBP80 and CBP20. The normalized SILAC ratios from this set (EDC over repressed) were divided by the normalized SILAC ratios from the previous set (active early over repressed). This generated abundance ratios for the proteins in the EDC relative to the active complex from which it forms. These data again indicated dramatic changes in composition in the transition between the two complexes.

Many proteins were depleted during formation of the EDC (*Figure 4C* left panel). PTBP1, RAVER1 and MBNL1, which were mostly lost between the repressed and active early complexes, declined even further in the EDC. Most hnRNP proteins, including hnRNPA1 and hnRNPA2B1, declined with the formation of the EDC. Interestingly, proteins such as hnRNP C, K, U and Q that had substantially increased in the active complex relative to the repressed complex, were now dramatically reduced in the EDC. Conversely and as expected, the U2 snRNP proteins are recruited during this transition (*Figure 4C* left panel and *Figure 3B* lane 3). Perhaps the most striking change upon EDC formation was a massive recruitment of SR proteins including SRSF3, 6, 7, 11, 12, and SR-related proteins, TRA2B (SFRS10), RBM39 and LUC7L2. TRA2B and SFRS3 increased 4-fold in the EDC over the active complex, while LUC7L2 went up 8-fold (*Figure 4C* left panel; *Supplementary file 1*). Again, the changes in hnRNP and SR proteins were verified by immunoblot (*Figure 4C* right panel), and we observed similar changes in the second MS experiment using PTBP1-depleted extract (*Supplementary file 1*). Together, the data indicate extensive remodeling of exon complexes both between the PTBP1-repressed and active early complexes and then in the transition from the active early exon complex to the EDC. Besides the binding of U2 during this latter transition, hnRNP proteins are removed and SR proteins are recruited.

## RAVER1 and MBNL1 are required for full repression of the N1 exon, while DDX5/17 and TRA2B activate splicing of the N1 exon

The early exon RNP complexes assembled in vitro are heterogeneous and not all proteins seen binding are expected to be key regulators of their assembly. To confirm the involvement of particular proteins in the splicing of the test exon in vivo, we performed RNAi knockdown experiments. These were performed on a HeLa cell line carrying an integrated splicing reporter that contained the same

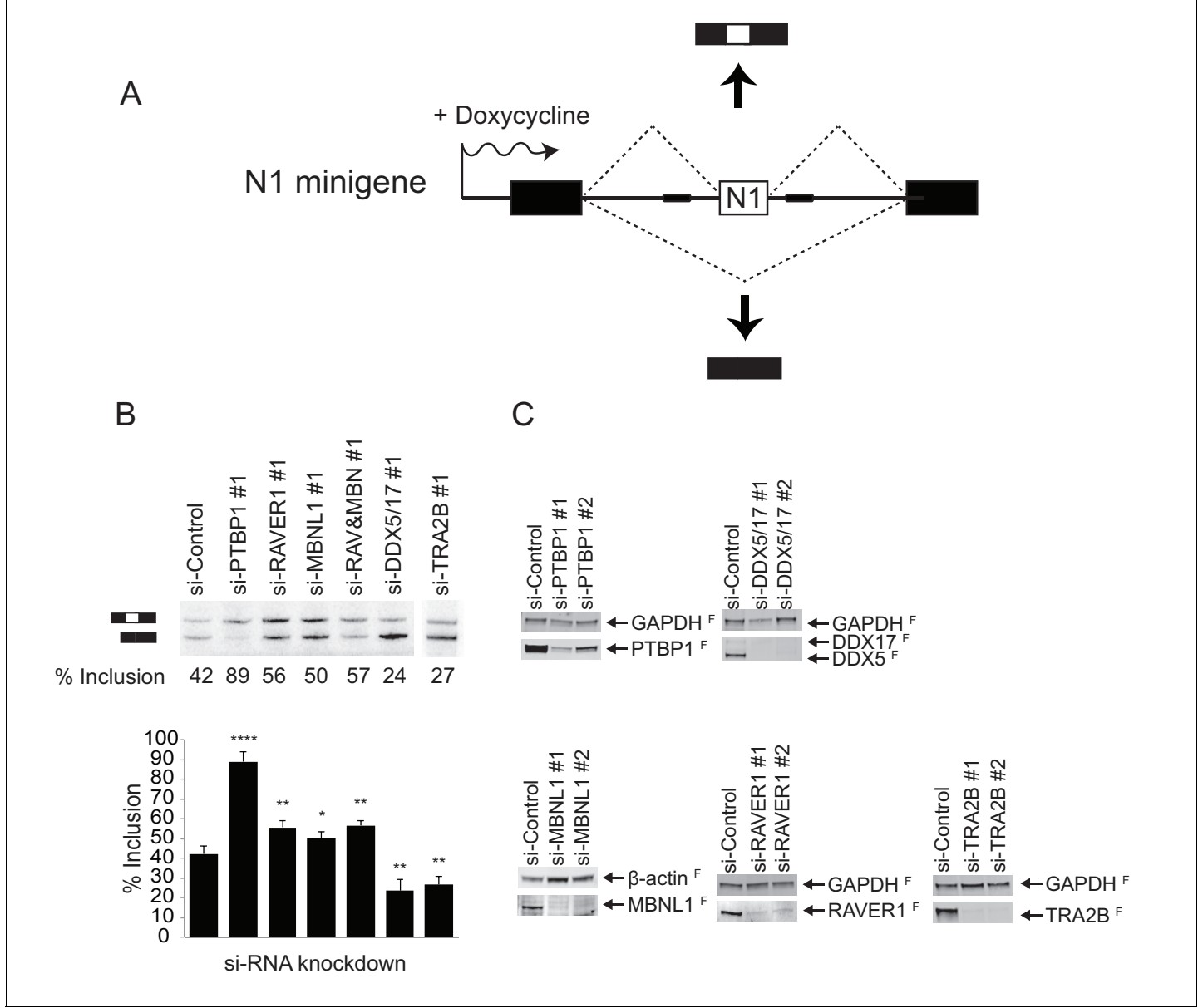

**Figure 5.** RAVER1 and MBNL1 are required for full N1 exon repression, while DDX5/17 and TRA2B activate N1 exon splicing. (**A**) Diagram of the minigene splicing reporter. The WT N1 exon was inserted into the intron between constitutive exons 1 and 2 of human β-globin (black boxes). The reporter was inserted into the Flp recombinase site of Flp-In HeLa cells (Life Technologies). Transcription of the reporter was induced by doxycycline. (**B**) SiRNA knockdown of splicing factors. The reporter cell line was transfected with the indicated siRNAs 48 hr before inducing reporter expression with doxycycline. Top panel: Total cellular RNA was harvested and assayed for reporter splicing by primer extension. Primer extension products were resolved on 4% urea-PAGE and autoradiographed. The exon-included and exon-skipped products are indicated, with the percent exon inclusion plotted below. Bottom panel: Averages of three experiments were plotted with standard deviations. p-values were calculated by comparing each knockdown to the control using a two-tail t-test. * = p<0.05; ** = p<0.01; **** = p<0.0001. (**C**) Immunoblots showing the efficiency of siRNA knockdown. Primer extension assays for siRNA set 2 is in *Figure 5—figure supplement 1*. GAPDH or β-actin served as loading controls. The superscripted 'F' indicates fluorescent secondary antibodies used for detection.

The following figure supplement is available for figure 5:

**Figure supplement 1.** A second set of siRNAs affects splicing similarly to the first set.

test exon flanked by β-globin exons and was induced with doxycycline (*Figure 5A*). We focused on factors that changed in each of the complexes analyzed. MBNL1 and RAVER1 are bound only in

conjunction with PTBP1 in the repressed complex. DDX5 and DDX17 bind in the active complex and are reduced as the EDC forms. TRA2B is increased in the active complex and highly enriched in the EDC. After RNAi knockdown and reporter induction, we isolated RNA from the cells and assayed exon inclusion by primer extension. Two siRNAs were tested for each protein and the efficiency of protein depletion was assessed by immunoblot (*Figure 5C*). In control siRNA-treated cells, the test exon exhibited ~40% inclusion in the reporter mRNA (*Figure 5B*). As expected, depletion of PTBP1 stimulated splicing of the test exon to ~90%. Depletion of MBNL1 or RAVER1 alone increased exon inclusion to 50 or 55% respectively, confirming both proteins as negative regulators of the test exon. The more limited effect from depleting these factors compared to PTBP1 may indicate the presence of other redundant repressor proteins or that PTBP1 is more important to maintaining exon repression than these cofactors. Double depletion of RAVER1 and MBNL1 together did not further increase splicing indicating that the factors are likely not redundant with each other.

Simultaneous depletion of DDX5 and DDX17 using an siRNA targeting both proteins (*Figure 5C*) sharply reduced the test exon splicing to ~25% (*Figure 5B*). Similarly, depletion of TRA2B (SRSF10) also significantly inhibited the test exon splicing to 27% (*Figure 5B*). Using a second set of siRNAs to deplete these proteins gave similar results (*Figure 5—figure supplement 1*). We also tested siRNAs targeting other proteins found in one or more of the complexes, including hnRNP C, DHX15, SRSF3 and LUC7L2, but the effects of depleting these proteins on splicing of the test exon were not as clear, perhaps due to the presence of paralogous proteins (data not shown). Collectively, these data confirm that RAVER1 and MBNL1 act as corepressors of N1 splicing with PTBP1, and that DDX5/17 and TRA2B act as activators of the N1 exon. The presence of the DDX5/17 proteins in the active complex indicates that these proteins may act during the remodeling events of an early exon complex into an EDC.

## Discussion

### Splicing Repression by PTBP1

Common models for the action of splicing regulators involve competitive binding at the splice sites (*Wagner and Garcia-Blanco, 2001*; *Singh et al., 1995*; *Valcárcel et al., 1993*; *Heiner et al., 2010*; *Zarnack et al., 2013*). However, PTBP1-repressed exons often have binding sites in the flanking introns distal to the target exon that are not expected to interfere with the binding of spliceosomal factors (*Linares et al., 2015*; *Xue et al., 2009*; *Llorian et al., 2010*; *Han et al., 2014*). Here, we show that in repressing such an exon, PTBP1 still allows U2AF and the U1 snRNP assembled onto an exon; however, further progression of the early exon RNP to form an EDC is inhibited (*Figure 6*).

When bound to sites flanking an exon, PTBP1 inactivated both its 5' and 3' splice sites (*Figure 1B and C*). We previously showed that the downstream PTBP1 makes physical contact with stem-loop 4 of the U1 snRNA bound at the 5' splice site upstream (*Sharma et al., 2011*). Although not proven, this PTBP1-U1 snRNP interaction may be sufficient to inactivate the 5' splice site by blocking an interaction between the repressed U1 and the U2 snRNP of the downstream exon complex (*Sharma et al., 2014*) . On the upstream side, it is possible that the bound PTBP1 allows normal U1-dependent U2AF assembly but blocks its further contact with the U2 snRNP. However, we find that the block in EDC assembly appears to be earlier than the final recruitment of the U2 snRNP.

We found dramatic differences in the early exon RNP complexes assembled in the presence or absence of PTBP1. The PTBP1-repressed complex contained several proteins whose binding was lost when PTBP1 was removed, either by binding site mutation or protein depletion (*Figure 4B*). The most notable changes were of MBNL1 and RAVER1, two proteins that are known to act as corepressors with PTBP1 on other exons (*Gromak et al., 2003*; *Gooding et al., 2013*). In RNAi depletion experiments, we find that RAVER1 and MBNL1 are both inhibitory for N1 splicing, although not as strongly as PTBP1. Work from the Smith and Curry labs has defined a precise interaction of a peptide in RAVER1 with RRM2 of PTBP1 (*Joshi et al., 2011*). The protein MATRIN3 has the same peptide and appears to make a PTBP1 interaction similar to RAVER1 (*Joshi et al., 2011*; *Coelho et al., 2015*). Although MATRIN3 is present in some of our complexes, its co-binding with PTBP1 was less clear than RAVER1 and MBNL1. The co-binding of RAVER1 and MBNL1 indicate that PTBP1 may nucleate the assembly of a larger inhibitory complex on the target exon. Early work by the

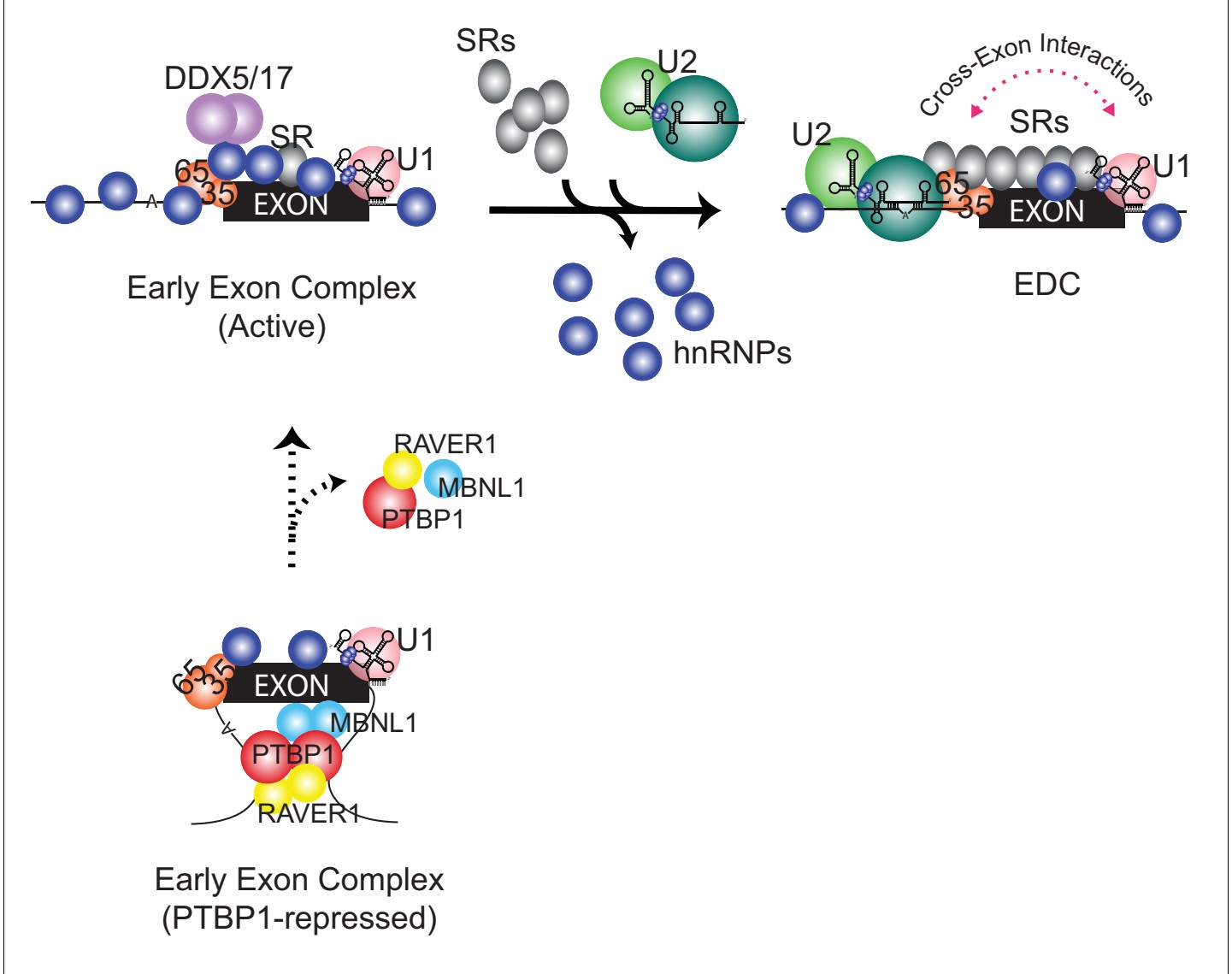

**Figure 6.** Compositional remodeling of a PTBP1-regulated exon RNP. The unrepressed exon is bound by abundant RNA-binding proteins including U2AF, the U1 snRNP, hnRNPs, and other factors to form the active early exon complex. This complex progresses to an exon definition complex with the loss of hnRNP proteins, the recruitment of SR proteins, and the assembly of the U2 snRNP at the branch point. The ATP-dependent RNA helicases DDX5 and DDX17 associate with the early exon complex and may contribute to the protein rearrangements. In the presence of PTBP1 (below) the exon is sequestered in a repressed state that still allows U2AF and U1 snRNP assembly but not the association of many early factors including hnRNPs, RNA helicases, and early bound SR proteins. This repressed exon complex cannot progress to the EDC.

Grabowski lab on GABA$_A$ receptor γ2 exon 10 also indicated that PTBP1 induced assembly of a larger inhibitory complex and prevented formation of an EDC-like complex (*Ashiya and Grabowski, 1997*). The components of that complex could not be analyzed and the effect of PTBP1 binding could not be uncoupled from competition with U2AF since their binding sites overlapped.

Other proteins that may be enriched in the repressed complex include the hnRNP F and H proteins and the protein CELF1, although these proteins showed less consistent differences among the three analyzed complexes. HnRNP H was previously shown to bind a silencer element within the exon itself, as well as bind to a downstream cluster of positive and negative regulatory elements (called the DCS) where its binding was dependent on PTBP1 (*Rooke et al., 2003*; *Markovtsov et al., 2000*; *Chou et al., 1999*; *Min et al., 1995*). The role of CELF proteins in N1 splicing has not been studied, but they are known to act either positively or negatively on the

splicing of other exons (*Lin et al., 2005*; *Ladd et al., 2001*; *Gromak et al., 2003*; *Charlet-B et al., 2002*; *Dembowski and Grabowski, 2009*). It will be interesting to further examine the roles of all these proteins.

Instead of directly blocking the binding of U2AF, PTBP1 may prevent assembly of other factors needed for U2 snRNP recruitment. In converse to RAVER1 and MBNL1, we found numerous factors that were absent from the repressed complex, but were recruited to the exon RNA if PTBP1 regulation was removed. Some of these are presumably needed for EDC assembly but bind much earlier than the U2 snRNP. We observed increased binding of the SR and SR-related proteins SFRS3 and TRA2B in the active early over the repressed complex. These proteins increased dramatically again in the EDC along numerous other SR proteins that were only observed in the EDC. Of interest, we found the ATP-dependent RNA helicases DDX5 (p68), DDX17 (p72) and DHX15 (PRP43) enriched in the active early complex. DDX5 and DDX17 showed substantial declines upon formation of the EDC, suggesting they act prior to its assembly and may be needed for remodeling the active early exon complex (see below) and are blocked from action by PTBP1. We are also interested in the role of ATP in active early complex assembly upon removal of PTBP1 repression. Determining this will require a more purified system, as we find that in crude extracts ATP alters the RNP assembled on nearly any RNA (data not shown).

Other splicing regulators are known to alter other steps of EDC assembly. As described above, PTBP1 bound within Fas exon 6 interfered with U1 snRNP-dependent recruitment of U2AF during exon definition, earlier than the action of PTBP1 studied here (*Izquierdo et al., 2005*). In contrast, House et al found that hnRNP L acted at a later step to prevent splicing of a CD45 exon through the hyper-stabilization of an assembled EDC containing the U1 and U2 snRNPs, and thus altering its transition to the full spliceosome (*House and Lynch, 2006*). Here, we found that even though U2AF was recruited, a step between U2AF binding and U2 snRNP recruitment was blocked by PTBP1 (*Figure 3C and D*). It will be very interesting to apply comparative proteomics to other exons in states such the hyper-stabilized and permissive EDCs. This may identify common factors targeted by regulation and further elucidate the process of exon definition.

## A pathway for exon definition complex assembly

Studies of spliceosome assembly have largely focused on the formation of the catalytic complex. Using short model introns from yeast or mammalian cells, these analyses have defined an intricate series of binding steps and conformational rearrangements leading to intron excision (*Will and Lührmann, 2011*; *Wahl et al., 2009*; *Hegele et al., 2012*). Relatively few studies have examined the exon complexes that are the necessary intermediate in the splicing of long metazoan transcripts (*Sharma et al., 2008*; *Schneider et al., 2010*). As the primary RNA transcript is synthesized, factors such as the hnRNP group of proteins and the U1 snRNP are thought to rapidly assemble with it (*Dreyfuss et al., 2002*; *Singh et al., 2015*). Some of these early recruited factors are required for the binding of subsequent spliceosomal components, including the U2 snRNP. Although the active early exon complex is heterogeneous in mobility, we found that it is fully bound by the known early factors U2AF and the U1 snRNP, as well as enriched for the DDX5/17 and TRA2B components needed for splicing, and it is efficiently converted to an EDC. This conversion requires more than the simple assembly of the U2 snRNP at the branch point sequence. The transition from the active early complex to an EDC was accompanied by a large-scale rearrangement of the exon RNP (*Figure 6*). It is likely that there are many as-yet-undescribed steps leading to the assembly of an exon definition complex.

When PTBP1-mediated repression was removed, many new proteins bound the exon to form the active early complex. Some of these early binding proteins may help recruit needed proteins, or they may force a particular sequence of assembly by hindering the binding of factors such as the U2 snRNP that need to bind later. Abundant hnRNP and other RNA-binding proteins assemble into this early exon complex but are then removed during the transition to the EDC (*Figure 4C*), including hnRNPs C, Q, R and K, as well as the residual PTBP1, RAVER1 and MBNL1.

Another striking change between the active early complex and the EDC is a massive recruitment of SR proteins. SR proteins were found to associate with the pre-mRNA and to stabilize the binding of the U1 snRNP and U2AF (*Staknis and Reed, 1994*; *Jamison et al., 1995*; *Wu and Maniatis, 1993*; *Kohtz et al., 1994*; *Krainer et al., 1990*). It is possible that these previously described interactions occur during the transition we observe after initial U1 snRNP and U2AF binding. On the

other hand, some SR proteins are present in the active early exon complex along with the U1 snRNP and U2AF. These early-bound SR proteins may serve to initially stabilize U1 and U2AF binding, before a more substantial recruitment of SR proteins during EDC assembly. During this later step, several SR and SR-like proteins are increased 3 to 4 fold, a similar enrichment to the U2 snRNP proteins, and may serve to bridge components bound at the 5' and 3' splice sites as previously proposed (*Fu and Ares, 2014*; *Long and Caceres, 2009*; *Busch and Hertel, 2012*). Understanding the roles of these SR and SR-like proteins needs further dissection. We are particularly interested in TRA2B, which increases three-fold in the active early complex and then an additional four-fold in the EDC, and whose depletion strongly inhibits exon inclusion in vivo (*Figure 5B*).

It seems likely that the loss of hnRNP proteins and gain of SR proteins during exon definition, are mediated by ATP-dependent RNA helicases, which are implicated in nearly all RNP assembly pathways (*Singh et al., 2015*; *Jarmoskaite and Russell, 2014*; *Cordin et al., 2012*; *Linder and Jankowsky, 2011*). The observed rearrangements during EDC formation require ATP, but the number of steps involving ATP hydrolysis is not clear. ATP is needed for U2 snRNP recruitment to the branch point through the action of UAP56 (SUB2) and the U2 snRNP subunit DDX46 (PRP5) (*Jarmoskaite and Russell, 2014*; *Cordin et al., 2012*). DDX46 is strongly recruited into our isolated EDC, but detection of UAP56 in our complexes was inconsistent. Given the timing of their recruitment, their association with other exon complexes, and their implication in the splicing of multiple alternative exons including the exon examined here (*Figure 5*), DDX5 (p68) and DDX17 (p72) are good candidates for driving new rearrangement steps during EDC assembly (*Sharma et al., 2008*; *Schneider et al., 2010*; *Kar et al., 2011*; *Dardenne et al., 2012*; *Dardenne et al., 2014*; *Camats et al., 2008*; *Guil et al., 2003*; *Liu, 2002*; *Lin et al., 2005*). DDX5 and DDX17 are extremely abundant in nuclear extracts and so far we have not achieved sufficient depletion of these proteins to test their roles in vitro. It will be very interesting to develop additional assays that can examine their function in more detail.

The remodeling of an exon complex is summarized in *Figure 6*. The assembly of PTBP1 and several corepressors forms the repressed complex that contains the U1 snRNP and U2AF but excludes numerous other factors from the exon RNP. Removal of PTBP1 allows formation of an active early complex that contains new hnRNP proteins as well as the RNA helicases DDX5 and DDX17 and some early-bound SR proteins. This early exon RNP is efficiently converted to the EDC with the loss of hnRNP proteins, the gain of SR proteins and the binding of the U2 snRNP. This sequence of events raises many questions. How does PTBP1 block assembly of so many different proteins? Does the initial binding of certain hnRNP proteins such as hnRNP C in the active early complex allow for recruitment of factors needed for EDC formation, or are they inhibitory? What drives the loss of hnRNP proteins and the gain of SR proteins during EDC formation? Is this the role of DDX5 and DDX17? Can smaller steps be discerned within these major assembly transitions? Given that exon definition appears to be the major pathway for recognizing splice sites in vertebrates, and a target for the regulation of splicing choices, it will be important to gain further understanding of EDC structure and assembly.

## Materials and methods

### Plasmid constructs and in vitro transcription

The plasmid pBS713 previously used by Sharma et al. containing the mouse Src N1 exon and flanking intron segments was modified using standard protocols to create pBS719WT containing a distal PTBP1-binding site upstream of the branch point and pBS719MUT containing a mutant upstream PTBP1-binding site (*Sharma et al., 2008*; *Amir-Ahmady et al., 2005*). Both plasmids contained the 3' splice site of Adenovirus Major Late first intron replacing the original N1 exon site. $^{32}$P-labeled RNA transcripts were synthesized using T7 RNA polymerase and gel purified as described previously (*Sharma et al., 2008*). Transcripts of the Adenovirus Major Late 5' and 3' exons were T7 transcribed from PCR amplified fragments of the pSPAd plasmid (*Solnick, 1985*). These 5' exon and 3' exon amplicons encompassed at least 50 bps into intron and exon sequences.

## Assembly of exon RNP complexes and native agarose gel analysis

The concentration of each RNA transcript was determined by UV absorbance on a Nanodrop spectrophotometer (Thermo Fisher Scientific, Waltham, Massachusetts). Exon RNP complexes were assembled in standard 20-µL standard splicing reactions containing 1 nM of RNA transcript, 2.2 mM $MgCl_2$, 0.4 mM ATP, 20 mM creatine phosphate, 1 U/µL RNAseOUT (Life Technologies, Carlsbad, California) and 60% v/v HeLa S3 nuclear extract (see below). Reactions were incubated at 30°C for 30 min, or as indicated. After assembly, heparin was added to the final concentration of 0.067 ug/uL, and incubation continued for 5 min. Reactions were resolved on 2.5% w/v agarose GTG gels (Lonza, Basel, Switzerland; in 25 mM Tris-Glycine pH 8.8) run at 100 volts for 3 hr at room temperature. Gels were fixed in 10% methanol-acetic acid for 30 min, and vacuum dried for 1 hr and with heat for 2 hr, before autoradiographed and scanned on a Typhoon phosphorimager (GE Healthcare, Chicago, Illinois). All biochemical analyses of RNP assembly, splicing, and protein composition were repeated 2 or more times to confirm the results.

## *Trans*-splicing

Unlabeled transcripts containing the N1 exon with wildtype or mutant PTBP1 sites, or the 5' and 3' Adeno Major Late (AML) exons were assembled into exon complexes in separate reactions under standard conditions for 20 min as above, except: the reaction volume was 10 µL and contained 2 nM of N1 exon substrate, or 50 nM of 5' or 3' AML exon substrates. After assembly, the reaction containing WT or MUT N1 exon was mixed with the 3' Adeno ML exon complex (for testing the N1 5' splice site) or with the 5' Adeno ML exon (for testing the N1 3' splice site). The combined reactions were supplemented with another dose of ATP and creatine phosphate, and incubated at 30°C for 3 hr. RNA was phenol extracted and ethanol precipitated from each reaction before assaying by primer extension analysis with a 5' $^{32}$P-labeled primer annealed to the 3' end of the *trans*-spliced product. Primer extension products were resolved on 8% 7.5 M urea PAGE and autoradiographed. Primer extension analyses of in vitro *trans*-splicing (*Figure 1*) and reporter gene splicing *in vivo* (*Figure 5*) were carried out as previously used by Carey et al, except that SuperScript III (Life Technologies, Carlsbad, California) was used and the incubation temperature was 50°C (*Carey et al., 2013*).

## Purification of exon RNP complexes

Exon complex purification was adapted from *Jurica et al. (2002)*. Before assembling in extract, WT and MUT exon RNAs containing three copies of the MS2 stem-loop at the 3' end were pre-incubated with a fusion protein of MS2 bacteriophage coat protein and maltose-binding protein (MS2-MBP; from Josep Vilardell) (*Macías et al., 2008*). The RNAs (10 nM) were assembled in large scale splicing reactions (400 µL) under standard conditions for 30 min. The reactions were chilled on ice for 5 min and spun at 20,000 g for 10 min at 4°C to remove large particulates. Heparin was omitted. Each reaction was layered onto a 15–45% w/v glycerol gradient (12 ml) prepared with 20 mM HEPES pH 7.9, 80 mM potassium glutamate, 2.2 mM $MgCl_2$ and 0.2 mM EDTA pH 8.0, on a BioComp gradient station (BioComp, New Brunswick, Canada). The gradients were spun in a SW41 rotor (Beckmann Coulter, Pasadena, California) at 37,000 rpm at 4°C for 15.5 hr, and fractionated into 25 fractions on the BioComp station. The abundance of complex in each fraction was determined by scintillation counting.

Fractions containing the relevant exon complexes were pooled and subjected to MBP affinity purification at 4°C. The pooled fractions were passed three times through amylose beads (NEB, Ipswich, Massachusetts) pre-equilibrated in wash buffer (20 mM HEPES pH 7.9, 80 mM potassium glutamate, 2.2 mM $MgCl_2$ and 0.2 mM EDTA pH 8.0). The beads were then washed with 20 column volumes of wash buffer. Exon complexes were eluted by incubating with one column volume of wash buffer containing 40 mM maltose at 4°C for 30 min. Yields of the eluted complexes were determined by scintillation counting against a known amount of RNA substrate.

## DNA-directed RNAse H Cleavage of U1 and U2 snRNPs

SnRNAs were targeted for RNAse H digestion with complementary DNA oligos (IDT, San Jose, California), targeting nucleotides 1–15 of U1 and U2 snRNA (*Black et al., 1985*). Oligos included, U1: CTGCCAGGTAAGTAT; U2: AGGCCGAGAAGCGAT; and a GAPDH targeted oligo as a negative control: GAGGTCAATGAAGGGGTCAT. RNAse H (NEB) cleavage conditions were as described

previously (*Merendino et al., 1999*). The treated nuclear extracts were either directly used for experiments or stored at −80°C.

## SYBR Gold staining

Urea PAGE gel was washed with 0.5x TBE buffer for 5 min twice to remove urea. SYBR Gold (Life Technologies) was diluted 1:10,000 in 0.5x TBE buffer and incubated with the gel for 10 min with gentle agitation. The gel was washed for 10 min with 0.5x TBE twice and scanned on a Typhoon phosphorimager using a 532 nm excitation laser and a 555 nm emission filter.

## Site-specific labeling, UV crosslinking and immuno-precipitation

Site-specific labeling of the N1 3' splice site was performed as a described previously (*Sharma et al., 2011*). The labeled substrates were incubated in standard splicing reactions for 10 min and chilled on ice before irradiating with 800 mJ of 254-nm UV in Stratalinker (Stratagene, San Diego, California) to crosslink proteins and RNA. 1 U/10 μL final concentration of RNAse T1 (Thermo Fisher Scientific) was added to the reaction and incubated at 37°C for 30 min. The samples were subjected to denaturing immuno-precipitation with mouse anti-U2AF65 (clone MC3, Sigma-Aldrich, St. Louis, Missouri) as described previously, except, 1x PBS buffer containing 0.05% NP-40 was used throughout the washing process (*Will et al., 2001*).

## SILAC and preparation of HeLa S3 nuclear extract

HeLa S3 cells (ATCC CCL-2.2) were a gift of Melissa Jurica. They were grown in SMEM medium (M0518, Life Technology). The medium was supplemented with 2 mg/mL sodium bicarbonate, 4 mM glutamate, 1X nonessential amino acids (Life Technologies), 5% PBS-dialyzed newborn calf serum (Omega Scientific, Tarzana, California), 1X penicillin–streptomycin (Life Technologies). For SILAC experiments, customized medium (similar to M8028 of Sigma-Aldrich without sodium bicarbonate, glutamate, arginine and lysine) was prepared by AthenaES, Halethorpe, Maryland. 13C-arginine (R6) and 13C-lysine (K6) were purchased from Cambridge Isotope Laboratories, Tewksbury, Massachusetts. The complete medium was reconstituted as above and with either 'light' (R0K0) or 'heavy' (R6K6) arginine and lysine at 12.6 g/L and 7.25 g/L, respectively. HeLa S3 cells were grown in either R0K0 or R6K6 for five to six doubling cycles to obtain high labeling efficiency (>99%). The cells were harvested in log phase and processed for nuclear extraction as described previously (*Black, 1992*; *Dignam, 1990*).

## Mass spectrometry

Protein mixtures were reduced, alkylated, and digested by the sequential addition of lys-C and trypsin proteases as previously described (*Kaiser and Wohlschlegel, 2005*). The digested peptide mixture was fractionated online using a 75-μM inner diameter fritted fused silica capillary column with a 5-μM pulled electrospray tip and packed in-house with 15 cm of Luna C18(2) 3-μM reversed-phase particles. Data was acquired on a Q-Exactive mass spectrometer (Thermo Fisher Scientific) (*Kelstrup et al., 2012*). Tandem MS (MS/MS) spectra were collected and subsequently analyzed using the ProLuCID and DTASelect algorithms (*Tabb et al., 2002*; *Xu et al., 2015*). Database searches were performed considering both the light and heavy versions of each peptide. Protein and peptide identifications were further filtered with a false positive rate of <5% as estimated by a decoy database strategy (*Elias and Gygi, 2010*). NSAF values were calculated as described (*Florens et al., 2006*). SILAC ratios for each peptide pair were determined using the Census algorithm (*Park et al., 2008*; *Park and Yates, 2010*).

## siRNA knockdown and RNA extraction

A reporter minigene, which contained the wild-type N1 exon from BS719WT flanked by constitutive exons 1 and 2 of human β-globin, was inserted into Flp-In T-REx HeLa cells (Life Technologies) to generate stable cell lines by following the manufacturer's protocol. The reporter was under the control of Tet repressor. After a stable pool of HeLa cells was established, cells were plated at $5 \times 10^5$ cells/6-cm plate in DMEM (Thermo Fisher Scientific) containing 10% FBS (Omega Scientific) and reverse transfected with siRNAs (sequences below) using RNAiMax (Life Technologies) according to the manufacturer's protocol. The final concentration of siRNA in the medium was 18.75 nM. The cells

were maintained at 37°C, 5% $CO_2$ for 48 hr before inducing reporter expression with 1 µg/mL doxy-cycline for eight hours, and then lysed with 1 mL of TRIzol (Life Technologies). To extract RNA, TRI-zol lysates were mixed with 1 volume of 100% ethanol before binding to a Direct-zol RNA miniprep column according to the manufacturer's protocol (Zymo Reseach, Irvine, California). One to three micrograms of total RNA in each knockdown was used for primer extension analysis of exon inclusion. Cellular splicing assays were repeated 3 times.

siRNA duplexes were previously reported or were designed using the online tool at http://sirna.wi.mit.edu. The duplexes were synthesized by Bioland Scientific, Paramount, California. The targeted sense strands are shown below.

si-Control : ACGUGACACGUUCGGAGAA
si-PTBP1 #1 : CAGUUUACCUGUUUUUAAA
si-PTBP1 #2 : UGACAAGAGCCGUGACUAC
si-RAVER1 #1 : UUGCCCAGCAGGUCCGACU
si-RAVER1 #2 : AGUCGGACCUGCUGGGCAA (*Chen et al., 2013*)
si-MBNL1 #1 : AACACGGAAUGUAAAUUUGCA (*Ho et al., 2004*)
si-MBNL1 #2 : CACUGGAAGUAUGUAGAGA
si-DDX5/17 #1 : GGCUAGAUGUGGAAGAUGU (*Dardenne et al., 2012*)
si-DDX5/17 #2 : AACCGCAACCAUUGACGCCAU (DDX5 3'UTR)
and AACAGCAGACUUAAUUACAUU (DDX17 3'UTR)
si-SFRS10 #1 : ACGCCAACACCAGGAAUUU (*Yang et al., 2015*)
si-SFRS10 #2 : GGAGGAUACAGAUCACGUU

## Immunoblot analysis and antibodies

Western blots were performed on total protein from siRNA knockdown cell cultures (56 hr post transfection) lysed in RIPA buffer supplemented with 1x cOmplete protease inhibitors (Roche, Basel, Switzerland) and Benzonase nuclease (Sigma-Aldrich). Lysates containing 40 ug of total protein were diluted in 5X SDS loading buffer, boiled for 5 min, and resolved on 8% Laemmli SDS PAGE. Proteins were transferred to PVDF membranes (EMD Millipore, Billerica, Massachusetts) that were blocked with 5% skim milk in PBS with 0.1% Tween 20 (PBST) and probed with primary antibodies overnight at 4°C. Membranes were washed in PBST and incubated with 5% skim milk containing Cy3- or Cy5-conjugated secondary antibody (Thermo Fisher Scientific; 1:3000 dilution) or HRP-conjugated secondary antibody (GE Healthcare; 1:5000 dilution). After washing, membranes were directly scanned on the Typhoon phosphorimager for fluorescence antibodies or developed with SuperSignal West Femto enhanced chemiluminescence (Thermo Fisher Scientific) for HRP-conjugated antibodies. Primary antibodies included: mouse anti-PTBP1 (clone BB7) (*Sharma et al., 2005*), rabbit anti-RAVER1 (A303-939A, Bethyl Laboratories, Montgomery, Texas), mouse anti-MBNL1 (LS-B4372, LifeSpan Biosciences, Seattle, Washington), rabbit anti-DDX5/17 (Genscript, China), mouse anti-phophorylated SR proteins (clone mAb104 (*Roth et al., 1991*). Note: anti-$I_gM$ secondary antibody must be used to probe mAb104), rabbit anti-SFRS10 (AV40528, Sigma-Aldrich), rabbit anti-hnRNPA1 (AN-351, Thermo Fisher Scientific), goat anti-hnRNPC1/C2 (sc-10037, Santa Cruz Biotechnology, Dallas, Texas), rabbit anti-hnRNPQ (LS-C31342, LifeSpan Biosciences), mouse anti-U2AF65 (clone MC3, Sigma-Aldrich), mouse anti-GAPDH (clone 6C5, Life Technologies), rabbit anti-β-actin (A2066, Sigma-Aldrich), mouse anti-U1-70K (clone CB7, [*Sharma et al., 2005*]) and rabbit anti-CBP80 (N0547-05B, US Biological, Salem, Massachusetts).

## Definition of PTBP1 binding sites on repressed exons

PTBP1-repressed exons in mouse ESC were identified in Linares et al (GEO: GSE71179) (*Linares et al., 2015*). Exons were chosen that both exhibited ≥15% increase in splicing after PTBP1 depletion by siRNA and contained nearby PTBP1 iCLIP clusters. Positions of iCLIP clusters were designated as in the upstream intron (a); in the BPS and polypyrimidine track (b); in the exon (c); or in the downstream intron (d), using the following criteria (*Supplementary file 1*). iCLIP clusters located more than 100 nts upstream from the 3'ss AG were designated 'a'; those in exons or downstream of the 5'ss were designated 'c' and 'd', respectively. When iCLIP clusters were located within 100 nts from the 3'ss, the intron sequences were subjected to BPS prediction as described by André et (*Corvelo et al., 2010*), to determine the potential BPS sites. If iCLIP clusters were located more than

25 nts upstream of the predicted BPS, they were designated 'a'; otherwise, they were designated 'b'. A list of PTBP1-repressed exons in mouse ESC with their PTBP1 binding site locations is shown in *Supplementary file 1*.

## Additional information

### Competing interests
DLB: Reviewing editor, *eLife*. The other authors declare that no competing interests exist.

### Funding

| Funder | Grant reference number | Author |
| --- | --- | --- |
| National Institute of General Medical Sciences | R01GM049662 | Douglas L Black |
| Howard Hughes Medical Institute | | Douglas L Black |
| National Cancer Institute | R21CA170786 | Shalini Sharma |
| National Institute of General Medical Sciences | R01GM089778 | James A Wohlschlegel |
| The Institute for the Promotion of Teaching Science and Technology, Thailand | Royal Thai Fellowship | Somsakul Pop Wongpalee |

The funders had no role in study design, data collection and interpretation, or the decision to submit the work for publication.

### Author contributions
SPW, Conception and design, Acquisition of data, Analysis and interpretation of data, Drafting or revising the article; AV, Acquisition of data, Analysis and interpretation of data; SS, Conception and design, Acquisition of data; DC, Acquisition of data, Analysis and interpretation of data, Drafting or revising the article; JAW, Conception and design, Acquisition of data, Analysis and interpretation of data; DLB, Conception and design, Analysis and interpretation of data, Drafting or revising the article

### Author ORCIDs
Douglas L Black, http://orcid.org/0000-0002-2705-8187

## Additional files

### Supplementary files
• Supplementary file 1. (a) Proteins present in the PTBP1-repressed (WT), the active (MUT) and the EDC complexes in SILAC experiment 1. (b) Proteins present in the PTBP1-repressed, the active-early and the EDC complexes in SILAC experiment 2. (c) Comparison of protein enrichments seen in SILAC experiment 1 and SILAC experiment 2. (d) NSAF scores of proteins identified in SILAC experiment 1. (e) NSAF scores of proteins from SILAC experiment 2. (f) Positions of PTBP1 crosslinking sites on validated PTBP1-repressed exons.

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
