## [Decision Letter]

Thank you for submitting your article "Large-scale remodeling of a repressed exon RNP to an exon definition complex active for splicing" for consideration by *eLife*. Your article has been reviewed by three peer reviewers, one of whom, Timothy W Nilsen (Reviewer #1), is a member of our Board of Reviewing Editors and the evaluation has been overseen by James Manley as the Senior Editor. Juan Valcárcel (Reviewer #2) has also agreed to reveal their identity.

As you will see all were generally quite positive about the work but all thought the paper would be strengthened by some experimental insight into the role(s) of DDX-17-5 in remodeling the inactive EDC into a splicing competent EDC. Please address this question as best as you can and also address minor comments as thoroughly as possible.

The reviewers have discussed the reviews with one another and the Reviewing Editor has drafted this decision to help you prepare a revised submission.

Reviewer #1:

Here Black and colleagues continue their investigation of how PTB can interfere with splicing. They focus on the role of PTB in preventing the formation of splicing competent Exon Definition Complexes (EDCs). In brief, using an imaginative array of constructs, and a variety of approaches (including SILAC) they demonstrate that active EDC complexes are formed through an elaborate series of protein recruitments and displacements that are independent of recruitment of either U1 snRNP or U2AF. It appears that hnRNP proteins are first recruited to the U1-U2AF complex and these are then displaced and replaced by SR proteins. The latter step in turn allows the recruitment of U2 snRNA and the formation of a splicing competent EDC. PTB, not bound to any critical cis-acting element, in conjunction with Raver and muscleblind, appears to inhibit this cascade at a very early step preceding the recruitment of hnRNP proteins. The work is clearly presented and the data quite clear. Overall the studies are in principle suitable for publication in *eLife*.

One of the more interesting aspects of the study is the requirement for DDX17-DDX5 for conversion of an inactive EDC to a splicing competent EDC. It would seem to be straightforward to show in knockdown cells what step these helicases perform. Such data would strengthen the paper (i.e. by immunoblot) and answer an obvious open question raised by the work.

Reviewer #2:

Black and colleagues present a detailed biochemical dissection of factors assembled on a PTB-repressed exon, on the exon harboring mutations in one of the PTB binding sites, or on the wild type exon upon biochemical depletion of PTBP1. The results indicate that 1) PTBP1 binding prevents the association of numerous hnRNP proteins, but not U1 snRNP nor the U2AF heterodimer, 2) when PTB binding is compromised, several hnRNP proteins assemble to form an early complex that, upon ATP-dependent remodeling, gives rise to an exon definition complex containing U2 snRNP as well as SR proteins and TRA2B.

These are interesting results from carefully performed experiments that shed light on the complex remodeling of RNPs involved in regulation of an exon. They can spur discussion and further work in other systems and represent a reference point for studies on the molecular mechanisms of splicing regulation, as illustrated by the discovery of the sequential assembly of hnRNP and SR proteins during de-repression of an exon.

I have no major criticism but I would recommend the authors to discuss the following points in a revised text, as detailed in the next section.

Reviewer #3:

This study by Black and colleagues describes a thorough analysis of the protein composition of a PTB-repressed complex, an early exon definition complex and a later exon definition complex. The trans-splicing system described is both elegant and robust and allows for an unprecedented identification of differences between these complexes. Strengths of this study are the quality of the data and the novelty and importance of characterizing such early complexes in spliceosome assembly. This is particularly timely given that we now have increasing detail regarding the later complexes in spliceosome assembly, but lack detail on earlier complexes where much of splicing regulation occurs. A short-coming of the study is that it is primarily descriptive, but the potential relevance of this data for other studies mitigates this concern – at least partially. It is true that the study raises more questions than it answers. The authors are well aware of this – they, in fact, list many of these questions at the end of the discussion. Answering any of these would increase the mechanistic insight of the study. Particular experiments that seem readily available given the system are:

1) The authors show that a few of the proteins identified in the mass spec studies impact the efficiency of splicing in cells. However, it is not shown that these proteins impact the stage of spliceosome assembly in which they were identified. The authors show that depletion of PTB is sufficient to alter early-to-EDC assembly. Does depletion of RAVER or MBNL do the same, or does depletion of DDX5/17 or Tra2 limit formation of the EDC?

2) Similarly, the author propose that widespread loss of hnRNPs and gain of SR proteins helps promote EDC formation, but this change in composition could be an effect not a cause of EDC formation. Experiments such as in point #1 with depletion of SR proteins would address this.

---

## [Author Response]

[…]

*Reviewer #1:*

*Here Black and colleagues continue their investigation of how PTB can interfere with splicing. They focus on the role of PTB in preventing the formation of splicing competent Exon Definition Complexes (EDCs). In brief, using an imaginative array of constructs, and a variety of approaches (including SILAC) they demonstrate that active EDC complexes are formed through an elaborate series of protein recruitments and displacements that are independent of recruitment of either U1 snRNP or U2AF. It appears that hnRNP proteins are first recruited to the U1-U2AF complex and these are then displaced and replaced by SR proteins. The latter step in turn allows the recruitment of U2 snRNA and the formation of a splicing competent EDC. PTB, not bound to any critical cis-acting element, in conjunction with Raver and muscleblind, appears to inhibit this cascade at a very early step preceding the recruitment of hnRNP proteins. The work is clearly presented and the data quite clear. Overall the studies are in principle suitable for publication in eLife.*

*One of the more interesting aspects of the study is the requirement for DDX17-DDX5 for conversion of an inactive EDC to a splicing competent EDC. It would seem to be straightforward to show in knockdown cells what step these helicases perform. Such data would strengthen the paper (i.e. by immunoblot) and answer an obvious open question raised by the work.*

We agree with the reviewer that it would strengthen the paper to have more data on this, and we had earlier made substantial efforts in this direction but without much success. Using both commercial and a custom made antibody, we have depleted DDX5/17 from nuclear extracts. These proteins are extremely abundant and the residual 15% of the protein makes the results difficult to interpret. The depleted extract did show a small decrease in EDC formation that could be restored with purified protein. However, these very small effects are difficult to draw conclusions from and we feel that it does not really strengthen the story to report on them. We now comment more clearly on the need for this type of confirmation.

With the encouragement of the reviewer, we have now also tried preparing mini nuclear extracts from HeLa cells depleted for DDX5/17 using a transfected si-RNA. Knockdown efficiency was about 90%, but unfortunately caused widespread cell death. The nuclear extract prepared from these cultures was blocked for EDC formation when compared to mini nuclear extracts prepared from control si-RNA treated cells. However, EDC assembly could not be rescued by bacterial or human cell purified recombinant DDX5/17. Thus the failure to assemble the EDC could result from many indirect effects of DDX5/17 depletion. Similar toxicity was observed when knocking down TRA2B.

[…]

*Reviewer #3:*

[…]

*1) The authors show that a few of the proteins identified in the mass spec studies impact the efficiency of splicing in cells. However, it is not shown that these proteins impact the stage of spliceosome assembly in which they were identified. The authors show that depletion of PTB is sufficient to alter early-to-EDC assembly. Does depletion of RAVER or MBNL do the same, or does depletion of DDX5/17 or Tra2 limit formation of the EDC?*

These are good experiments but they have proven technically quite difficult due to inadequate antibodies, the presence of paralogs or other factors. We have made particular efforts to examine the function of DDX5/17 as described in the response to reviewer 1 above. Unfortunately, we will need to develop new methods to address these questions.

*2) Similarly, the author propose that widespread loss of hnRNPs and gain of SR proteins helps promote EDC formation, but this change in composition could be an effect not a cause of EDC formation. Experiments such as in point #1 with depletion of SR proteins would address this.*

This has similar limitations as above but with more paralogs to be dealt with. As far as we are aware, immunodepletion of the complete set of SR proteins with antibodies such as Mab104 has not been successful. This leaves methods such as Magnesium precipitation for SR depletion, which is not very specific and difficult to control. We have a small amount of preliminary data showing that the addition of SR proteins (from ammonium sulfate precipitates) can enhance EDC formation in a splicing-deficient nuclear extract, but it is not yet possible to relate this to the Early-Active to EDC transition we see here.